# In Silico Conformational Features of Botulinum Toxins A1 and E1 According to Intraluminal Acidification

**DOI:** 10.3390/toxins14090644

**Published:** 2022-09-17

**Authors:** Grazia Cottone, Letizia Chiodo, Luca Maragliano, Michel-Robert Popoff, Christine Rasetti-Escargueil, Emmanuel Lemichez, Thérèse E. Malliavin

**Affiliations:** 1Department of Physics and Chemistry Emilio Segré, University of Palermo, Viale delle Scienze, 90128 Palermo, Italy; 2Department of Engineering, University Campus Bio-Medico of Rome, Via Á. del Portillo 21, 00128 Rome, Italy; 3Department of Life and Environmental Sciences, Polytechnic University of Marche, Via Brecce Bianche, 60131 Ancona, Italy; 4Center for Synaptic Neuroscience and Technology, Istituto Italiano di Tecnologia, Largo Rosanna Benzi 10, 16132 Genova, Italy; 5Institut Pasteur, Université Paris Cité, CNRS UMR6047, Inserm U1306, Unité des Toxines Bactériennes, 75015 Paris, France; 6Institut Pasteur, Université Paris Cité, CNRS UMR3528, Unité de Bioinformatique Structurale, 75015 Paris, France; 7Laboratoire de Physique et Chimie Théoriques (LPCT), CNRS UMR7019, University of Lorraine, 54506 Vandoeuvre-lès-Nancy, France; 8Laboratoire International Associé, CNRS and University of Illinois at Urbana-Champaign, 54506 Vandoeuvre-lès-Nancy, France

**Keywords:** *Clostridium botulinum*, botulinum toxin, molecular dynamics, residue protonation, homology modeling

## Abstract

Although botulinum neurotoxins (BoNTs) are among the most toxic compounds found in nature, their molecular mechanism of action is far from being elucidated. A key event is the conformational transition due to acidification of the interior of synaptic vesicles, leading to translocation of the BoNT catalytic domain into the neuronal cytosol. To investigate these conformational variations, homology modeling and atomistic simulations are combined to explore the internal dynamics of the sub-types BoNT/A1 (the most-used sub-type in medical applications) and BoNT/E1 (the most kinetically efficient sub-type). This first simulation study of di-chain BoNTs in closed and open states considers the effects of both neutral and acidic pH. The conformational mobility is driven by domain displacements of the ganglioside-binding site in the receptor binding domain, the translocation domain (HC${}_{\mathrm{NT}}$) switch, and the belt α-helix, which present multiple conformations, depending on the primary sequence and the pH. Fluctuations of the belt α-helix are observed for closed conformations of the toxins and at acidic pH, while patches of more solvent-accessible residues appear under the same conditions in the core translocation domain HC${}_{\mathrm{NT}}$. These findings suggest that, during translocation, the higher mobility of the belt could be transmitted to HC${}_{\mathrm{NT}}$, leading to the favorable interaction of HC${}_{\mathrm{NT}}$ residues with the non-polar membrane environment.

## 1. Introduction

The botulinum neurotoxins (BoNTs), produced by *Clostridium botulinum*, are among the most powerful toxic compounds found in nature, provoking typical deadly flaccid paralysis of the host [1]. BoNTs are traditionally classified into seven serotypes, termed A–G [2], as well as the more recently discovered H (or FA), J, and X serotypes [3,4]. Among them, the BoNT/A1 sub-type is the most-used toxin in medical applications. Despite the fact that botulinum neurotoxin serotypes A–G inhibit acetylcholine release, they hijack different neuronal receptors, cleave different intracellular components of the Soluble N-ethylmaleimide-sensitive-factor Attachment protein Receptor (SNARE) machinery (which underpins the fusion of neurotransmitter-containing vesicles), and exhibit different neuron intoxication and intracellular stability kinetics [5,6]; in particular, BoNT/A1 and BoNT/E1 display quite different properties.

As for their structural architecture, the proteolytically activated BoNTs are formed by two protein chains connected by one disulfide bridge: the light chain (LC) and the heavy chain (HC). Over the past fifteen years, numerous X-ray crystallographic structures of BoNTs, obtained from samples prepared as single-protein chains, have been determined [7,8,9]. Although these structures present conformational variations, they all display similar domain organization. Figure 1A provides an overview of this organization, using the BoNT/A1 structure as an example [7]. In the figure, the LC (green color) contains the catalytic site, whereas HC is composed of two distinct domains: an N-terminal translocation domain HC${}_{N}$ (≃50 kDa) responsible for the LC delivery into the cytosol, and a C-terminal domain (HC${}_{C}$) (≃50 kDa) responsible for receptor binding. HC${}_{N}$ spans the belt (cyan) and the core translocation domains (HC${}_{\mathrm{NT}}$, orange), whereas HC${}_{C}$ spans the N- and C-terminal receptor binding domains (HC${}_{\mathrm{CN}}$, magenta; HC${}_{\mathrm{CC}}$, red). In more detail, the catalytic domain displays an α–β fold. The translocation domain HC${}_{\mathrm{NT}}$ is composed of a bundle of α-helices and loops. HC${}_{\mathrm{CN}}$ contains predominantly β-sheets arranged into a jelly-roll motif, while HC${}_{\mathrm{CC}}$ folds into a β-trefoil. For sake of clarity, we denote the extremity of HC${}_{\mathrm{NT}}$ located closest to the disulfide bridge, marked with an asterisk in Figure 1A, as the top of HC${}_{\mathrm{NT}}$; the opposite extremity being the bottom of HC${}_{\mathrm{NT}}$. The two long α-helices of HC${}_{\mathrm{NT}}$ are denoted by helix 1 and helix 2 (Figure 1B). Two other sub-domains of HC${}_{\mathrm{NT}}$ are the HC${}_{\mathrm{NT}}$ switch, formed by three α-helices and located in the middle of HC${}_{\mathrm{NT}}$, and the HC${}_{\mathrm{NT}}$ C-terminal α-helix on the other side of HC${}_{\mathrm{NT}}$ (Figure 1C). It should be noted that the C-terminal α-helix, present in the BoNT/A1 structure [7], is unfolded in the X-ray crystallographic structure of BoNT/E1 [9].

Furthermore, the X-ray crystallographic structures display two distinct conformations—termed open and closed—characterized by different arrangements of the LC and receptor binding domains, with respect to the central HC${}_{\mathrm{NT}}$ helical domain. In the open conformation, the LC and receptor binding domains are far apart, presenting as open wings lying on either side of HC${}_{\mathrm{NT}}$ (see Figure 1A). In the closed conformation, the LC and receptor binding domains come into close contact, like “closed” wings of a butterfly [10]. The open conformation has been observed in most of the X-ray structures [7], whereas the closed conformation has only been observed in the structure of BoNT/E1 [9]. Some of the X-ray crystallographic structures have been determined at acidic pH values in the range 4–6 [8], but no structural variation has been observed. Interestingly, BoNTs are closely related to the tetanus toxin TeNT, which presents a closed conformation with different organization, when compared to BoNT/E1 [11,12]. BoNT/A associates with the non-toxic non-hemagglutinin (NTNH) protein at acidic pH, forming stable complexes resistant to protease and acidic degradation [13]. Investigation of the BoNT/A-NTNH assembly by small angle X-ray scattering (SAXS) has revealed BoNT/A conformational intermediates between open and closed ones [14].

**Figure 1 toxins-14-00644-f001:**
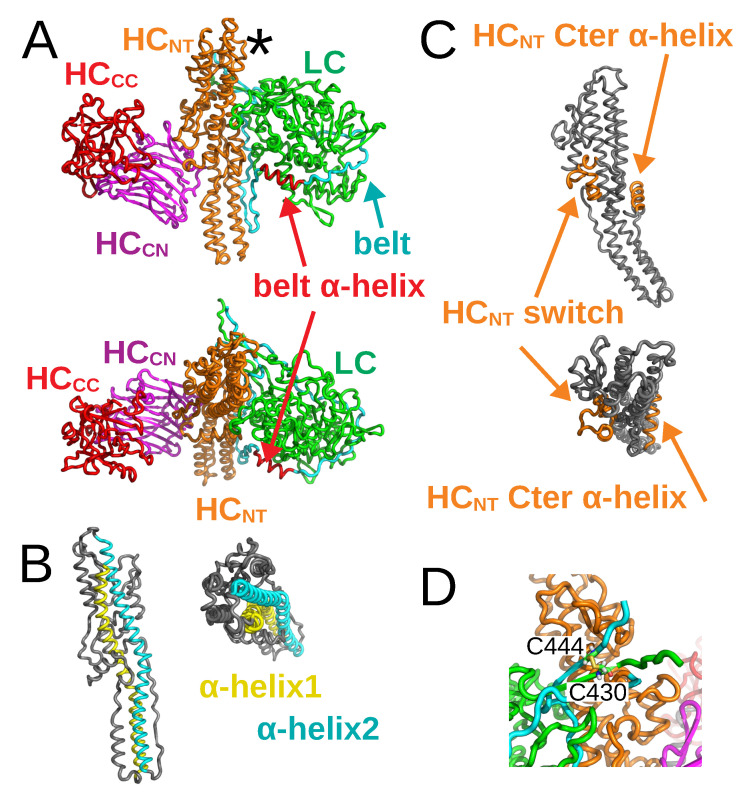
X-ray crystallographic structure of BoNT/A1 in open state, drawn in cartoon with various domains in different colors (PDB entry: 3BTA [7]). (**A**) Full view of the structure with domains LC (green), belt (cyan), HC${}_{\mathrm{NT}}$ (orange), HC${}_{\mathrm{CN}}$ (magenta), and HC${}_{\mathrm{CC}}$ (red). The belt α-helix is colored in red. The top of the HC${}_{\mathrm{NT}}$ domain, close to the disulfide bridge, is indicated with an asterisk. Top: Side view. Bottom: Upper view. (**B**) Translocation domain in BoNT/A1 structure with α-helix 1 (cyan) and -helix 2 (yellow). Left: Side view. Right: Upper view. (**C**) Translocation domain in BoNT/A1 structure with the HC${}_{\mathrm{NT}}$ switch and the C-terminal α-helix in orange. Top: Side view. Bottom: Upper view. (**D**) Close-up of the disulfide bridge between C${}^{430}$ and C${}^{444}$, connecting the two chains.

As for their functional mechanism, when approaching the terminal button of target neurons, BoNTs recognize two distinct classes of receptors [15]: complex gangliosides [16], specifically expressed on vertebrate pre-synaptic neuronal membranes; and synaptic vesicle SV2 (for BoNT/A1 and BoNT/E1) [17,18] or synaptotagmin for (BoNT/B) [19]. BoNT/A1 is capable of utilizing all three SV2s (A,B,C), whereas BoNT/E1 uses only SV2A or SV2B, but not SV2C; at least not in cultured neurons [18]. Receptor binding by BoNTs is followed by endocytosis of the toxin within recycling synaptic vesicles. Following the step of endocytosis, pH acidification within the vesicle interior triggers toxin-mediated translocation of LC through the endosomal membrane, using a mechanism whose molecular basis is not yet fully understood [20,21,22]. Once BoNTs–LC–Zn${}^{2+}$-metalloproteases are delivered into the cytosol of neuron pre-synaptic endings, they cleave a protein of the SNARE complex [23]. As the SNARE complex constitutes the central component of calcium influx-mediated fusion of synaptic vesicles for neurotransmitter release, their cleavage in cholinergic neurons by LC-BoNTs leads to deadly flaccid paralysis [24].

In the comparison between BoNT/A1 and E1, the difference recorded in the onset and duration of paralysis between these two sub-types covers contrasting behaviors at the level of the various physiological steps underlying the mechanisms of action of these two neurotoxins. Notably, after injection of botulinum neurotoxins BoNT/A and BoNT/E into the muscle of patients, the neuromuscular junction recovers more rapidly from the paralytic effect of BoNT/E than that of BoNT/A [5]. It has also been reported that BoNT/E LC is degraded more rapidly by the ubiquitin–proteasome system in the cytosol as compared to BoNT/A LC, which is relatively stable [25]. The increased stability of BoNT/A LC involves the activity of two debiquitinating enzymes: VCIP135, which prevents BoNT/A LC degradation by the proteasome, and USP9X, which prevents its lysosomal degradation [26]. The stabilized form of BoNT/A co-localizes with SNARE components at the pre-synaptic membrane, while BoNT/E LC localizes into the cytosol [27,28]. Moreover, the translocation of the catalytic domain of BoNT/A from the acidified lumen of endosomal compartments to the cytosol has been shown to be slow, relative to that of BoNT/E [29].

These findings indicate the importance of capturing the molecular dynamics of BoNT translocation across endosomal membranes. In this respect, some internal dynamics have been described in recent studies on BoNTs and TeNT. First, a SAXS study of TeNT has shown that, under acidic pH, the gyration radius (R${}_{G}$) decreases—an observation which might be related to the appearance of a closed conformation [11]. In addition, a region of HC${}_{\mathrm{NT}}$, named the HC${}_{\mathrm{NT}}$-switch, has been shown to display conformational transitions enabling membrane insertion of the translocation domain [30]. A recent analysis of BoNT/B and BoNT/E structures by Cryo-electron microscopy (Cryo-EM) [31] has demonstrated that these structures are more mobile than the corresponding X-ray crystallographic structures. Although some of these structures might be not completely functional, these observations point to an internal mobility of BoNTs, compared with the relative structure uniformity of BoNTs recorded by X-ray crystallography.

Molecular-level information on the internal mobility of toxins can be obtained through atomistic simulations [32], which can provide insight into the different possible conformations of BoNTs, as well as how they are affected by the environment. Few molecular dynamics (MD) simulations of BoNTs have been reported. A study has reported on the full-length BoNT/A in water at different pH and temperatures [33], while another study has investigated the interaction of the BoNT/A receptor binding domain with the synaptic vesicle protein 2C (SV2) luminal domain [34]. A recent study [35] has focused on the pH-dependent structural changes of BoNT/E1, based on extensive MD simulations at various pH values and on SAXS analysis.

In the present work, we explore the internal dynamics of two full-length BoNTs (BoNT/A1 and BoNT/E1) in a large water system, along a time scale of hundreds of nanoseconds, using the information provided by X-ray crystallographic structures along with homology modeling. The initial conformations of BoNT/A1 and E1 were constructed, in order to analyze the behaviors of the open and closed states of these toxins, in both neutral and acidic pH conditions. We investigate the protein internal dynamics at ternary and quaternary levels, as well as relate the observed conformational changes to relevant physiological steps. A general mechanism is proposed for the initiation of translocation in BoNTs, and a comparative analysis of the cleaved toxins BoNT/A1 and E1 provides a first-level description of putative structural determinants for the different translocation kinetics driven by intraluminal acidification in these two BoNT serotypes.

## 2. Results

In this section, we present the results obtained through analyzing several descriptors of the protein structure and dynamics along the MD trajectories (Table 1), recorded starting from cleaved X-ray crystallographic conformations or from trans models, as described in detail in Section 4.1. Here, the term ’trans models’ refers to structures fully obtained by homology modeling calculations (e.g., open conformations of E1 and closed conformations of A1), to be distinguished from conformations modeled based upon the available experimental X-ray structures (e.g., closed E1 and open A1 conformations).

We started with an analysis of residue protonation, which reflects both the pH and the particular BoNT conformation (open or closed). Residue protonation, defined at the modeling stage, has a direct effect on the protein internal dynamics, determining intra- and inter-domain long-range interactions. The Root Mean Square Deviation (RMSD) between conformations, as well as the distances between domains, were utilized to obtain global information on the structural rearrangements in BoNTs with time. Atomic Root Mean Square Fluctuations (RMSFs) were analyzed to define local motions in distinct domains, while monitoring of inter- and intra-domains hydrogen bonds provided information complementing the global structural results. We then focused on the dynamics of three domains: the belt, the core translocation domain, and the binding receptor domains.

The definitions of the BoNT domains utilized in this work are reported in Table 2.

### 2.1. Analysis of Protonation at Varying pH in Different States

The protonation of amino-acid residues under various states and at different pH values are listed in Appendix A. In addition to the three protonation states of histidines—namely, protonated histidines on Nδ (HSD), protonated histidines on Nϵ (HSE), and doubly protonated histidines (HSP)—the other protonated residues were glutamate (GLU) and aspartate (ASP), in which a hydrogen was added to the side-chain carboxyl group.

Different numbers of protonated residues were observed in the BoNT domains. A large number of protonated residues, in the range of 7–13 for BoNT/A1 and 6–12 for BoNT/E1, were observed in LC. The number of protonated residues increased from around 8 at neutral pH to 10–13 at acidic pH; in this condition, the number was larger in the closed conformations of BoNT/A1 (around 13) than the open ones. Another large cluster of protonated residues, including from 9 to 14 residues, was located at acidic pH in the domain HC${}_{\mathrm{NT}}$. As with LC, this number increased with acidic pH and was larger for the closed conformations. Some of these residues were located in the HC${}_{\mathrm{NT}}$ switch (D${}^{640}$, E${}^{656}$ in A1clo47r and E${}^{616}$, E${}^{623}$, E${}^{632}$ in E1ope47), while others (D${}^{838}$ in A1clo47r, D${}^{829}$ in E1clo47 and E1ope47) were close to the C-terminal helix of HC${}_{\mathrm{NT}}$. In BoNT/E1, other residues are located close to the α-helix 1 and -helix 2; in particular, to residues K${}^{775}$, E${}^{636}$, E${}^{677}$ in E1clo47 and E${}^{773}$, E${}^{781}$ in E1ope47 and E1ope47r. Overall, few residues were protonated in the belt and in the domains HC${}_{\mathrm{CN}}$ and HC${}_{\mathrm{CC}}$, except for the open state of BoNT/E1 at acidic pH (E1ope47 and E1ope47r).

### 2.2. Variations of the Intra- and Inter-Domain Organization in BoNTs

The combined analysis of RMSD, RMSFs, and hydrogen bond networks provided useful insights for tertiary and quaternary modifications in different conformational and protonation states.

The RMSD of Cα atoms along the MD trajectories (Figure 2) showed plateaus after only 50 ns in some cases. Plateau values were up to 7 Å for the whole structure (top plots), with a temporary jump up to 10 Å at around 125 ns in the trajectory E1clo70 (green curve).

Very flat and low profiles around 2-4 Å were observed for the light chain (LC, bottom row), whereas the RMSD calculated on the heavy chain (HC, middle row) dominated the total RMSD values. The RMSD values for LC were in the range 2–4 Å for A1 whereas, for E1, some RMSD curves were smaller than 2 Å, and others were in the range of 3–4 Å. The LC, composed mostly of the catalytic domain, thus displayed a stable conformation.

The BoNT/A1 trajectories starting from a trans model displayed RMSD values in the 4–6 Å range (green, orange, olive green, and brown curves). These values were slightly larger than those observed for the open state trajectories generated from X-ray cleaved models (magenta and cyan curves). A similar feature was observed for the BoNT/E1, with a larger gap between cleaved X-ray (orange and green curves) and trans models (blue, pink, magenta and cyan curves). In these cases, the use of additional restraints (described in Section 4.1 and Appendix A) to enforce the interaction between the belt α-helix and its environment (pink and blue curves) did not actually reduce the RMSD.

The trajectories starting from X-ray cleaved models (A1ope47, A1ope70, E1clo47, E1clo70) displayed different behaviors in A1 and E1. Indeed, unlike BoNT/A1, the trajectory E1clo70 (green curve) displayed a large jump around 125 ns, in which the receptor-binding domains (HC${}_{\mathrm{CN}}$ and HC${}_{\mathrm{CC}}$) and the LC domain moved slightly apart (Appendix A).

This lack of stability may have arisen bias in the X-ray crystallographic structure (3FFZ) induced by the crystal packing, or by the use of a unique protein chain to produce the sample for crystallographic purposes.

Nevertheless, protonation effects could possibly play a role. Indeed, in analogy with the conformational transition observed experimentally at an acidic pH value of 5.0 for TeNT [11], one may conceive that the open form of A1 at pH 7.0 (magenta curve) would be more stable than at pH 4.7 (cyan curve); meanwhile, to the contrary, the closed form of E1 would be more stable at pH 4.7 (orange curve) than at pH 7.0 (green curve). Actually, this is what was observed for the HC RMSD in Figure 2, comparing the A1 magenta and cyan curves (A1ope70 vs. A1ope47) and the E1 orange and green curves (E1clo47 vs. E1clo70), respectively. To resume, two factors contributed to the protein internal stability: artifacts of the homology modeling templates, as well as protonation effects, which shift the equilibrium toward one or the other conformation of the protein.

The distributions of Cα RMSD calculated for the individual BoNT domains, are displayed as box-plots in Figure 3. We found that the RMSD values—except for those of belt and HC${}_{\mathrm{CC}}$—were clustered within a much narrower range (1–4 Å for the medium values) than the global RMSD values observed in Figure 2. This observation supports a model of mobility in which the individual domains fluctuate around stable conformations, while most of the motions of the overall structure arise from relative displacements of the domains. This agrees with previous results from molecular dynamics simulations on BoNTs [33,35]. In this frame, the outlier values observed for the belt in the closed state of BoNT/A1 are not surprising, as this region is an extended loop connecting the catalytic and HC${}_{\mathrm{NT}}$ domains. One should also notice that HC${}_{\mathrm{CC}}$ obtained much larger RMSD values than the other domains, particularly in the closed state of BoNT/A1. Meanwhile, LC, HC${}_{\mathrm{NT}}$, and HC${}_{\mathrm{CN}}$ presented smaller RMSD values.

A repeated feature in Figure 3 is the increase in RMSD value for MD trajectories starting from trans models, with respect to those starting from cleaved X-ray models. This was previously observed for global RMSD (Figure 2), and is related to the percentage of identity in the 35–45% range between the primary sequences of the two toxins. Nevertheless, some exceptions were observed, such as the belt and HC${}_{\mathrm{CC}}$ displaying similar RMSD values in all E1 trajectories.

RMSFs along the LC and HC residues (Figure 4) were analogous for BoNT/A1 and BoNT/E1, with peaks observed at similar positions. Two peaks were observed at the two extremities of the domain HC${}_{\mathrm{NT}}$, for residues 746–751 (A1) and 723–734 (E1) (peak a) located in the loop at the top of HC${}_{\mathrm{NT}}$ (indicated by asterisk in Figure 1A), and for residues 813–826 (A1) and 798–805 (E1) (peak b) located in the loop at the bottom of HC${}_{\mathrm{NT}}$. Another peak was observed in HC${}_{\mathrm{NT}}$ for the residues 632–656 (A1) and 601–656 (E1) (peak c), located in the HC${}_{\mathrm{NT}}$ switch and in the linker connecting the HC${}_{\mathrm{NT}}$ switch and the α-helix 2. The domain HC${}_{\mathrm{CC}}$ displayed numerous peaks of mobility for both toxins; the largest was observed for A1clo70r (indicated with the letter d).

LC displayed only two peaks in the RMSF profiles, with values larger than 3 Å. Peak 1, located around residues 63–65 for A1 and 53–60 for E1, corresponds to a loop on the top of the catalytic domain. Peak 2, located around residues 393–394 for A1 and 392–399 for E1, corresponds to a region just before the disulfide bridge.

Distances between the geometric centers of several domains were monitored (Figure 5) along the MD trajectories. The distances between the LC and HC${}_{\mathrm{NT}}$ domains and between HC${}_{\mathrm{CN}}$ and HC${}_{\mathrm{CC}}$ displayed similar values across closed and open forms, cleaved X-ray models or trans models, and with pH changes. On the other hand, the distances between HC${}_{\mathrm{NT}}$ and HC${}_{\mathrm{CN}}$ or HC${}_{\mathrm{CC}}$ domains increased when comparing the open and closed conformations, regardless of whether the starting point was a cleaved X-ray model or a trans model. This increase differed between the two toxins: BoNT/A1 displayed a continuous increase, whereas BoNT/E1 displayed a jump. The moving apart of HC${}_{\mathrm{CN}}$ and HC${}_{\mathrm{CC}}$ from the translocation domain might have a functional meaning. Indeed, BoNTs initially interact with receptors through the HC${}_{\mathrm{CC}}$ domain for BoNT/A1, and through the HC${}_{\mathrm{CN}}$ and HC${}_{\mathrm{CC}}$ domains for BoNT/E1 [36], then translocate through the vesicle endosomal membrane. Moving apart HC${}_{\mathrm{CN}}$ and HC${}_{\mathrm{CC}}$ from the HC${}_{\mathrm{NT}}$ domain could free HC${}_{\mathrm{NT}}$ and the catalytic domains, making them available for translocation through the vesicle membrane. In this picture, the closed state induced by acidic pH [11] would be the most prone to translocation. In addition, by assuming that the increase in the HC${}_{\mathrm{NT}}$/HC${}_{\mathrm{CN}}$ and HC${}_{\mathrm{NT}}$/HC${}_{\mathrm{CC}}$ distances is an early event required for translocation, the steepest distance jump observed in E1 could support, at a molecular level, the experimental observation that pH-dependent translocation of BoNT/E is faster, relative to that of BoNT/A [29].

Long-range hydrogen bonds were detected along the trajectories using the Python package MDAnalysis [37,38]. The number of hydrogen bonds present more than 60% of the time and involving residues separated by more than 10 residues in the sequence was calculated between and within BoNT domains (Figure 6). The number of long-range hydrogen bonds within the LC domain (and, to a lesser extent, within HC${}_{\mathrm{CN}}$) was high in all trajectories (displaying however a slightly decrease in trans models), in agreement with the smaller RMSD values observed for LC and HC${}_{\mathrm{CN}}$ (Figure 3). In contrast, the number of hydrogen bonds between the belt and LC domains, and within HC${}_{\mathrm{NT}}$ and HC${}_{\mathrm{CC}}$, was mostly smaller and decreased for trajectories starting from trans models. The smaller number of hydrogen bonds within HC${}_{\mathrm{CC}}$ was in agreement with the large RMSD (Figure 3) and RMSF (Figure 4) values obtained for this domain.

### 2.3. Flexibility of the Lipid- and Ganglioside-Binding Domains in HC${}_{\mathrm{CC}}$

Analysis of the root-mean-square fluctuation profiles revealed large internal mobility in the domain HC${}_{\mathrm{CC}}$ (Figure 4). In particular, a peak (labeled *d*) was observed for loop 1188–1198 in the domain HC${}_{\mathrm{CC}}$ for the trajectory A1clo70r, which was much larger than any such peak observed for E1. In the initial conformation of HC${}_{\mathrm{CC}}$, the loop 1188–1198, which in A1 corresponds to the lipid-binding loop (LBL), adopted a β-strand conformation, forming a small β-sheet with the β-strand spanning residues 1252 to 1254 (sequence VAS) close to and partially overlapping the stretch of residues 1254–1257 (SNWY). Similarly, the corresponding loop in E1 (residues 1164–1179) established in the initial conformation of HC${}_{\mathrm{CC}}$ was a β sheet with the stretch of residues VAS (residues 1216–1218) close to the stretch of residues STWYY (1218–1222). Remarkably, the stretch SXWY is the conserved ganglioside-binding site (GBS), which was identified first in tetanus neurotoxin, as well as in BoNT/A, B, E, F, and G [39,40,41]. Interestingly, in E1 GBS, key interacting residues unique to BoNT/E have been identified, along with a significant rearrangement of loop 1228–1237, upon carbohydrate binding [42]. In BoNT/B, the LBL is located between the GBS and the receptor binding site, and is significantly exposed to the solvent; in particular the side-chains of W${}^{1248}$ and W${}^{1249}$ [43]. In the X-ray crystallographic structures of BoNTs, the regions LBL and GBS are close in 3D space, and correspond to well-defined binding regions (see, e.g., Appendix A for A1 in the open state; PDB entry: 3BTA).

The distributions of distances corresponding to backbone hydrogen bonds between the two β strands initially present in the X-ray crystallographic structures were monitored along MD trajectories (Figure 7). In the open state of A1, the interactions between β strands 1188–1198 and 1252–1254 established initially were maintained during the trajectory (see Appendix A). In all other trajectories, the interaction between β strands was lost. In general, the mobility observed in this region during simulations was more pronounced in the case when additional restraints (Appendix A) were applied on the α helix, particularly for the closed state in A1 (A1clo70r, olive green box) and for the open state of E1 (E1ope70r, pink box).

The mobility of the loop LBL in BoNT/A1 and E1 may seem paradoxical, as the X-ray crystallographic structures determined so far have shown quite well-defined interaction pockets with protein receptor and gangliosides [44]. Overall, the very mobile domain HC${}_{\mathrm{CC}}$ observed here points to a different interaction mechanism than that inferred from the X-ray crystallographic structures. Indeed, a recent closer investigation of the interaction between BoNT/B, ganglioside GT1b, and synaptotagmin has revealed [44] that a complex GT1b-synaptotagmin exists prior to BoNT/B binding, stabilizing the conformation of the synaptotagmin juxtracellular domain, in a way quite different to that observed in the structures. The high flexibility of the HC${}_{\mathrm{CC}}$ domain observed in our models is consistent with this variability of BoNT conformations and their interactions with receptors. Nevertheless, it should be noted that we did not investigate the binding of BoNTs with N-glycans in the present study.

### 2.4. Conformational Variations in the Core Translocation and Belt Domains

To investigate the mobility of the core translocation and belt domains, several structural descriptors were analyzed, the combination of which may help in dissecting the possible earliest events in the translocation. In particular, we analyzed the solvent-accessible residue surfaces, the behavior of the backbone dihedrals in the belt domain and in the HC${}_{\mathrm{NT}}$ switch region, as well as the bending of the α helices 1 and 2 in HC${}_{\mathrm{NT}}$.

**Figure 7 toxins-14-00644-f007:**
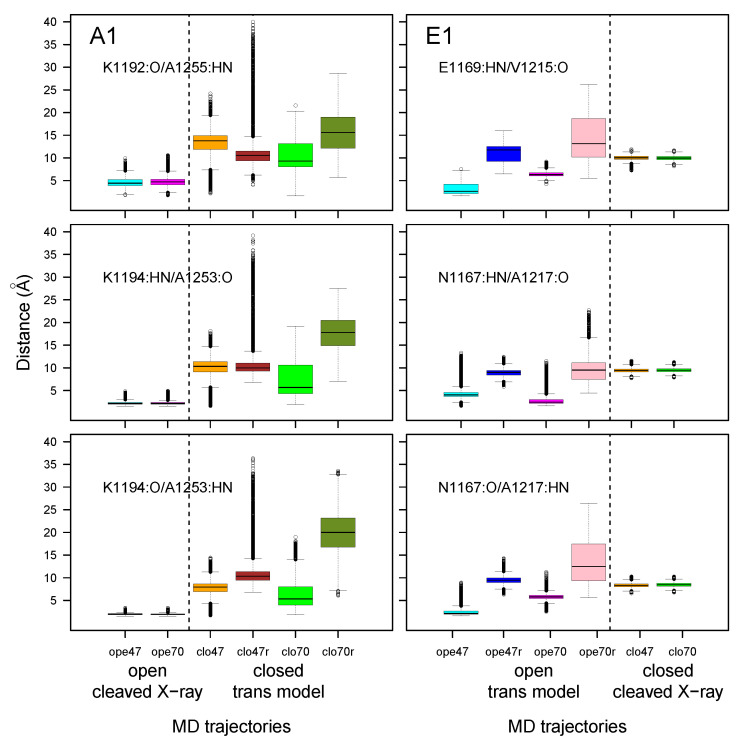
Box-and-whisker plots representation of distributions of distances corresponding to the hydrogen bonds between the amide hydrogens (HN) and carbonyl oxygens (O) of residues 1188–1198 (LBL) and 1252–1254 (close to the GBS) for A1 and of residues 1164–1179 (LBL) and 1216–1218 (close to the GBS) for E1. The color code for the boxes is as follows: cyan (A1ope47, E1ope47), blue (E1ope47r), magenta (A1ope70, E1ope70), pink (E1ope70r), orange (A1clo47, E1clo47), brown (A1clo47r), green (A1clo70, E1clo70), and olive green (A1clo70r). The dashed lines mark the separation between open and closed states; see Figure 3.

#### 2.4.1. Solvent-Accessible Residue Surface

The solvent-accessible surfaces of residues were calculated along the trajectories and the time-averaged values, clustered according to criteria detailed in the Section 4. We grouped the residues into four sets: (i) Residues more accessible in closed than in open state at both pH; (ii) residues more accessible in open state than in closed state at both pH; (iii) residues more accessible in closed state at pH 7.0 than in other states; and (iv) residues more accessible in closed state at pH 4.7 than in other states. Residues belonging to the four sets are listed in Table 3. Their positions along the protein sequence are shown in Appendix A. To convey the information where patches of more exposed residues are located, in the different sub-types and pH conditions, residues are displayed in Figure 8, on the structures of the open conformations of BoNT/A1 and BoNT/E1.

Overall, for both BoNTs, clusters of green residues (i.e., more accessible in closed than in open conformations) were observed in the translocation domain (HC${}_{\mathrm{NT}}$) and, to a lesser extent, in the catalytic domain (LC). To the contrary, the residues colored in cyan (more accessible in the open than in the closed conformation) were mostly scattered along the sequence or in the 3D structure.

In detail, residues more accessible in the closed state were located in specific regions of HC${}_{\mathrm{NT}}$, as shown in Appendix A. In BoNT/A1, L${}^{843}$ and I${}^{863}$ (Table 3) are located at the two extremities of the C-terminal α-helix connecting HC${}_{\mathrm{NT}}$ and HC${}_{\mathrm{CN}}$ (Figure 1C), the residue V${}^{783}$ is behind the HC${}_{\mathrm{NT}}$ switch, and the residues K${}^{796}$ and E${}^{799}$ are in the α-helix 2 toward the top of HC${}_{\mathrm{NT}}$. In BoNT/E1, L${}^{624}$ is located in the HC${}_{\mathrm{NT}}$ switch; E${}^{761}$, I${}^{764}$, and S${}^{763}$ in helix 2; W${}^{686}$ is in helix 1; and the residues K${}^{769}$, N${}^{772}$, E${}^{773}$, K${}^{775}$, I${}^{776}$ and N${}^{777}$ are located toward the bottom of α-helix 2. These accessible residues observed in the HC${}_{\mathrm{NT}}$ domain can be related to previous experimental observations [45] indicating that, in BoNT/A LC-HCT, residues located in an α-helix close to the bottom extremity of HC${}_{\mathrm{NT}}$ displayed an increase in fluorescence intensity and blue shift to 530 nm when I830C-NBD bound to liposomes, in agreement with the transfer into a non-polar environment. Remarkably, among all of the systems investigated, the number of solvent-exposed residues in the HC${}_{\mathrm{NT}}$ domain was the largest in the closed state of E1. These results support the hypothesis of Kumaran et al. [9] on the faster translocation of BoNT/E [29]; which, according to the authors, could be related to the exposure of one side of the translocation domain to the solvent.

**Figure 8 toxins-14-00644-f008:**
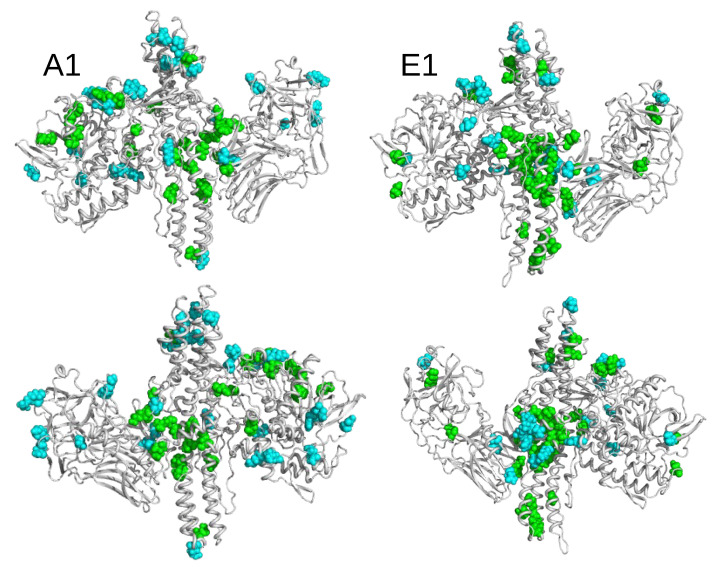
Open conformations of BoNT/A1 and BoNT/E1 drawn in cartoon and displayed in two opposite orientations. The residues displaying variations of accessible surfaces are shown in van der Waals representation, and colored according to the following: Green residues, more accessible in closed than in open state; cyan, more accessible in open than in closed state.

Among the residues displaying changes in accessible surface (Table 3), several residues changed protonation states (Appendix A) along with a change in pH: H${}^{39}$, H${}^{170}$ (LC), E${}^{799}$ (HC${}_{\mathrm{NT}}$), and D${}^{866}$ (HC${}_{\mathrm{CN}}$) in A1; and E${}^{741}$, E${}^{773}$ (HC${}_{\mathrm{NT}}$), and D${}^{866}$ (HC${}_{\mathrm{CN}}$) in BoNT/E1. In the LC domain, some residues protonated at acidic pH were located in the neighborhood of several residues changing accessible surfaces (Appendix A). Indeed, in BoNT/A1, the residues R${}^{48}$, T${}^{52}$, I${}^{154}$, and N${}^{205}$, which were more accessible in closed than in open state, are respectively located in the neighborhood of residues H${}^{39}$, D${}^{58}$, E${}^{164}$, and D${}^{513}$, which are protonated in closed state at acidic pH. In BoNT/E1, the residues S${}^{292}$, L${}^{391}$, and R${}^{394}$, which were more accessible in closed state, are located close to H${}^{125}$ and E${}^{189}$, which are protonated at acidic pH. As higher accessibility as well as protonation may facilitate membrane interaction, their occurrences in residues close in 3D space could also induce a co-operative effect in the interaction.

In BoNT/A1, several residues of LC were more accessible in the closed state at acidic pH; namely, F${}^{36}$, I${}^{111}$, P${}^{116}$, P${}^{156}$, H${}^{170}$, G${}^{179}$, T${}^{183}$, A${}^{228}$, L${}^{232}$, Y${}^{233}$, Y${}^{251}$, F${}^{290}$ ], and K${}^{320}$. Given that most of these are hydrophobic residues, their exposure to solvent could be related to loosening of the LC domain fold (Appendix A). This may also affect the active site, as A${}^{228}$, L${}^{232}$, and Y${}^{233}$ are close to the residues H${}^{223}$ and H${}^{227}$ of the catalytic site, belonging to the same helix-spanning residues (217–233). Such disruption of the LC tertiary structure was also suggested by the high value of the LC RMSD in closed form at acidic pH (Figure 3, upper left panel), as well as the reduction of intra-protein hydrogen bonds, with respect to the open form (Figure 6, upper left panel). The loosening of the LC fold has been observed experimentally at acidic pH [46], which could serve as the first step in preparing for its subsequent translocation through the vesicle membrane.

#### 2.4.2. Internal Mobility of Belt α Helix and HC${}_{\mathrm{NT}}$ Switch

In order to examine the variations of conformations in the belt domain and HC${}_{\mathrm{NT}}$ switch, the circular variances V(ϕ) and V(ψ) [47] of the backbone dihedral angles ϕ and ψ were calculated (Equation (Equation 1) in Materials and Methods). In the center of the belt, the α-helices 485–496 (A1) and 465–471 (E1) observed in the X-ray crystallographic structures (Figure 1A) displayed minimum values of V(ϕ) and V(ψ) for most of the trajectories, whereas peaks of fluctuations were located mostly in the flanking outside regions (Figure 9). The open state of A1 presented the largest interval of null values, corresponding to the most stable α helix. Overall, the closed states displayed shorter ranges of minimal circular variances in the α-helix. For closed states (orange, green, brown, and olive green curves), as well as for E1ope70 and E1ope47 (magenta and cyan curves), acidification induced the appearance of peaks inside and outside of the α-helix. This increased belt mobility could be a starting point for belt destabilization at acidic pH before translocation.

The circular variance calculated on the HC${}_{\mathrm{NT}}$ switch (Figure 10) most often displayed a peak of mobility in the middle of this region, spanning residues 635–640 and 648–652 for BoNT/A1 and 610–618 for BoNT/E1. For the two BoNTs, the region of maximum variance included the loop between the helices $\alpha_{A}$ and $\alpha_{B}$ of the HC${}_{\mathrm{NT}}$ switch, the names of these helices having been proposed by Lam et al. [30]. This loop was also the region displaying the largest conformational transition in [30].

The circular variance in HC${}_{\mathrm{NT}}$ switch presented different features under the various trajectory conditions: In the closed state of BoNT/A1, the mobility increased at acidic pH, whereas it decreased at acidic pH for the open state of BoNT/A1. In BoNT/E1, no pH effect was observed.

**Figure 9 toxins-14-00644-f009:**
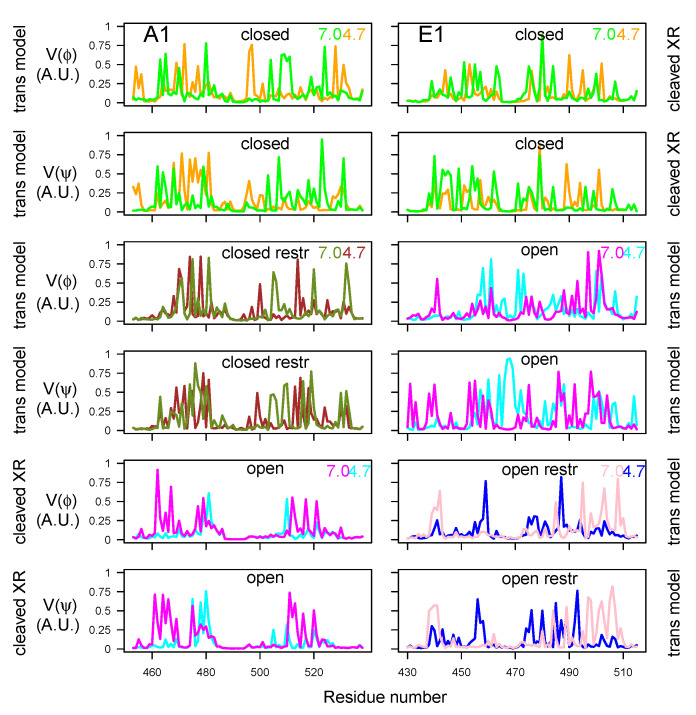
Variations of circular variances V(ϕ) and V(ψ) (Equation (Equation 1)) [47] calculated over the 150–300 ns interval of the trajectories, for dihedral angles ϕ and ψ of the belt domain. The color code is as follows: cyan (A1ope47, E1ope47), blue (E1ope47r), magenta (A1ope70, E1ope70), pink (E1ope70r), orange (A1clo47, E1clo47), brown (A1clo47r), green (A1clo70, E1clo70), and olive green (A1clo70r). The pH values used to define the protonation level of residues are written on the right part of the plots. The title of each plot refers to the BoNT type (A1/E1), as well as the conformational state (open/closed). Thus, the titles “open restr” and “closed restr” correspond to the trajectories of A1clo47r, A1clo70r, E1ope47r and E1ope70r, in which restraints were used during the homology modeling step (Appendix A).

**Figure 10 toxins-14-00644-f010:**
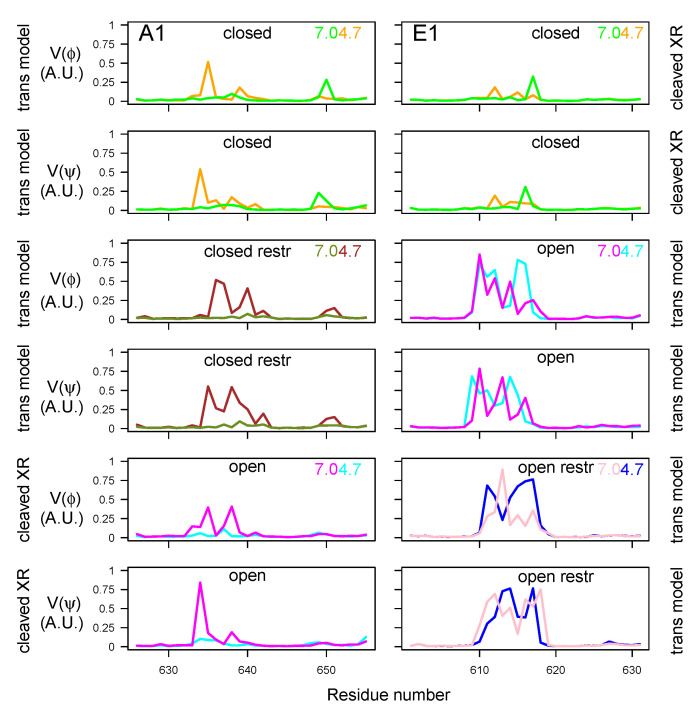
Variations of circular variances V(ϕ) and V(ψ) (Equation (Equation 1)) [47] calculated on the 150–300 ns interval of the trajectories, for dihedral angles ϕ and ψ of the switch domain in HC${}_{\mathrm{NT}}$. The color code, as well as the titles and annotations, are the same as in Figure 9.

#### 2.4.3. Hc${}_{\mathrm{NT}}$ Helix Bending

The bending of α helices located in HC${}_{\mathrm{NT}}$ was analyzed using Bendix [48] (see Figure 11). In the α-helices 1 and 2, the maximal bending angles were located in the residue ranges 705–715 and 777–797 (A1) and 698–718 and 737–798 (E1). This corresponded to a bend already observed in the X-ray structures and initial models, located at the bottom of the HC${}_{\mathrm{NT}}$ switch. The local bending angles of α-helix 1, spanning residues 680–740 (A1) and 660–720 (E1), displayed similar profiles under all conditions, with peaks of bending at the middle of the helix (Figure 11). On the other hand, those of helix 2, spanning residues 760–820 (A1) and 740–800 (E1), presented variations, depending on the closed or open state and on the type of BoNT. In BoNT/E1, the bending peaks were located around residue K${}^{775}$, close to several protonated residues (see Section 2.1).

To summarize, complicated correlation patterns were observed between the destabilization of the α-helix belt, the more solvent-accessible residue surfaces in HC${}_{\mathrm{NT}}$, the residue protonation, the bending of HC${}_{\mathrm{NT}}$ helices, and the internal mobility of the HC${}_{\mathrm{NT}}$ switch. In several cases, at acidic pH and in closed states, more accessible surfaces as well as flexibility in the belt and HC${}_{\mathrm{NT}}$ switch were observed simultaneously. The observation of larger accessible surfaces at acidic pH was in agreement with a recent molecular dynamics study on BoNT/E1 [35]. The results of the present work are discussed in the following section.

**Figure 11 toxins-14-00644-f011:**
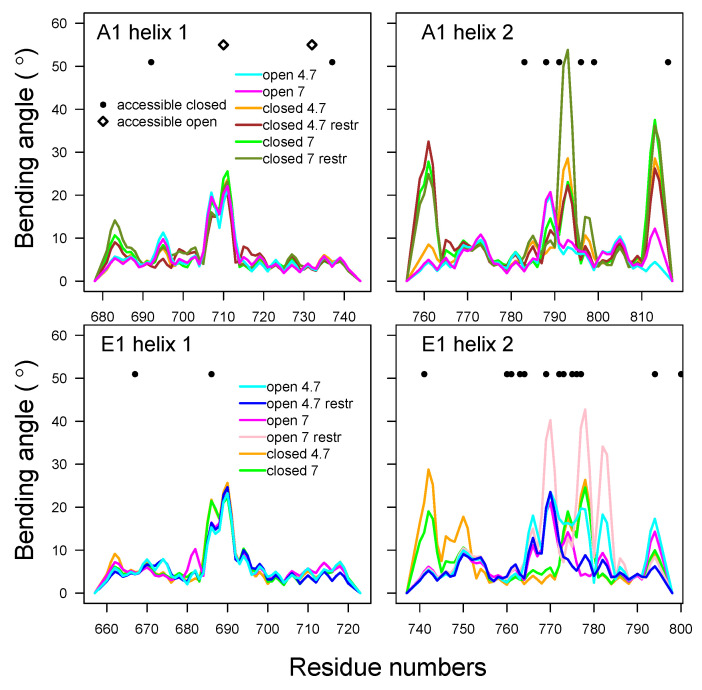
Analysis of the bending angles of the helices in the domain of translocation of BoNTs A1 and E1. Local bending angles were calculated using the Bendix VMD plugin [48], on the α-helices 678–744 and 756–817 in A1, and 657–723 and 737–798 in E1. The profiles of these angles are drawn with the color curves, coded as follows: cyan (A1ope47, E1ope47), blue (E1ope47r), magenta (A1ope70, E1ope70), pink (E1ope70r), orange (A1clo47, E1clo47), brown (A1clo47r), green (A1clo70, E1clo70), and olive green (A1clo70r). In the same plots, residues that present a difference in solvent-accessible surface are indicated by different symbols, depending on whether the accessible surface is larger in the trajectory starting from closed (•) or open (⋄) conformations.

## 3. Discussion

The relevance of BoNTs proteins in the medical field has stimulated the formulation of mechanistic hypotheses accounting for variations in their kinetics and stability in recent years. However, the sequence of events at the basis of their function is far from being understood at the molecular level, although numerous structures of the interaction partners have been determined. Some key unanswered questions about BoNTs are as follows: First, how do the ternary and quaternary structural changes of the protein relate to relevant physiological processes? Second, can we capture the structural signatures underlying pH-induced mechanisms? Third, could molecular-level knowledge on the action of different BoNTs guide structure-based engineering for toxin-based therapeutics?

Addressing these points first requires reliable, atomically refined structures of the protein complexes, as well as extensive knowledge on the network of residue–residue interactions which, unfortunately, is currently available as X-ray or CryoEM maps only for selected BoNT sub-types; in particular, for conformations engineered as single chains. In any event, the protein structure–dynamics–function paradigm requires the knowledge of how proteins dynamically behave in water and/or in the presence of a membrane.

The aim of this work was to provide insight into the different possible conformations of BoNTs, as well as how they are affected by the environment, through the use of computational tools. Two BoNT sub-types—A1 and E1—were studied using full atomistic molecular dynamics simulations, corresponding to a cumulative trajectory duration of 3.6 μs, in both open and closed conformations. Two different protonation states were considered, corresponding to acidic and neutral pH values (i.e., 4.7 and 7.0, respectively).

Different initial structures were exploited, starting from cleaved X-ray crystallographic conformations, as well as from trans models based on sequence alignment between BoNT/A1 and BoNT/E1. Given that the two studied toxins present a sequence identity of about 35–45%, depending on the chain, we cannot affirm that the trans models correspond to conformations significantly populated in the actual conformational landscape.

The obtained results provide a structural and functional annotation of full-length BoNTs composed of two distinct protein chains, which is in agreement with a recent molecular dynamics study of BoNT/E1 at various pH values [35]. A global overview of the simulation results indicates that the global movement of BoNTs is dominated by the relative motions of the domains, in agreement with the results of Chen et al. [33] on BoNT/A. The parallel use of different starting points allows for the detection of conformational features which can be related to the BoNT functions, such as movement of domains; the internal flexibility of the HC${}_{\mathrm{CC}}$ ganglioside-binding site, of the HC${}_{\mathrm{NT}}$ switch, and of the belt; and higher solvent accessibility of residues in the HC${}_{\mathrm{NT}}$ and LC domains. Moreover, the data pointed out connections between different regions, such as the belt and the HC${}_{\mathrm{NT}}$ domains, or the HC${}_{\mathrm{NT}}$ domain and the HC${}_{\mathrm{CN}}$ and HC${}_{\mathrm{CC}}$ domains. Given that most of these observations can be related to independent experimental observations, they provide insight into the functional dynamics of BoNTs. In particular, residues displaying larger accessible surfaces in the translocation domain HC${}_{\mathrm{NT}}$ could be starting anchors for the interaction of BoNTs with the membrane.

Remarkably, the BoNT/E1, when simulated at pH 7, displayed a large divergence from the starting X-ray crystallographic structure. This supports a picture of the BoNT/E1 structure in solution, in which the LC and HC${}_{\mathrm{CN}}$/HC${}_{\mathrm{CC}}$ domains spontaneously move away from each other. One should also note that HC/E takes various positions, with respect to LC, in the structures of the *C. botulinum* progenitor M complex of type E [49,50]; this agrees with an internal mobility of BoNT/E1 which allows it to occupy conformations different from the closed one.

The variations of residue protonation due to pH had some effects, although being of minor extent when compared with the results of experiments conducted in the presence of membrane [20,21,51]. At acidic pH and in closed state, a patch of HC${}_{\mathrm{NT}}$ residues close in 3D space was more exposed to the solvent. The observation of this patch correlates with the largest number of non-histidine residues protonated in HC${}_{\mathrm{NT}}$. The internal mobility of the belt (and, in particular, of the belt α-helix) allows one to propose a model for the initiation of translocation, in which the higher mobility of belt is transmitted to HC${}_{\mathrm{NT}}$ through the connection loop, inducing a more favorable interaction of HC${}_{\mathrm{NT}}$ residues with the non-polar membrane environment.

A recent article has described structures of BoNT/B and BoNT/E, obtained by Cryo-electron microscopy (Cryo-EM) [31]. Several observations reported were in good agreement with the data presented here. First, the main structural variations seemed to derive from overall movement of the domains with respect to each other, similar to the observations made here, where the RMSD values of individual domains (Figure 3) were smaller than the global RMSD values (Figure 2). The binding domain in BoNT/B shifted up to 2 Å around HC${}_{\mathrm{CN}}$; this seemed to be further accentuated at the HC${}_{\mathrm{CC}}$ domain [31]. This observation is in agreement with the increased distance between the domains HC${}_{\mathrm{NT}}$ and HC${}_{\mathrm{CN}}$/HC${}_{\mathrm{CC}}$ observed here (Figure 5). In addition, the EM map quality around the binding domains HC${}_{\mathrm{CN}}$ and HC${}_{\mathrm{CC}}$ was generally weaker, compared to the rest of the toxin, and the map was particularly well-defined for LC, in agreement with the RMSD and RMSF profiles observed in MD trajectories (Figure 2 and Figure 4), as well as the mobility of the lipid-binding loop (LBL). In the cryo-EM map of BoNT/B, the belt was well-ordered, except for a small surface-exposed α-helix; namely, the one for which we observed variations in internal mobility (Figure 9). More generally, the results of the present study confirm the fact that the X-ray crystallographic structures of BoNTs, determined from samples formed from one chain, do not completely capture the structural features of the toxin in solution, which plays an essential role in the physico-chemical aspects underlying their functional physiological processes. This observation is in agreement with that of a recent work [44], showing that BoNT/B in interaction with receptors seems to display much higher molecular flexibility than that deduced from the X-ray crystallographic structures of BoNTs.

## 4. Materials and Methods

### 4.1. Preparation of Starting Conformations of Toxins

In order to prepare systems in which BoNTs are formed of two proteins only connected by a disulfide bridge, the structures from PDB entries 3BTA (A1) [7] and 3FFZ (E1) [9] were cleaved, eliminating the peptides ^438^TKSLDKGYNK^447^ and ^420^GIR^422^; the numbering being taken from PDB files 3BTA and 3FFZ (MR Popoff, personal communication). Consequently, the BoNT domains are defined, in the present work, according to the ranges of residues given in Table 2. After removing the cleaved sequences, the disulfide bridge was established between C^430^ and C^444^ in BoNT/A1 (Figure 1D), and between C^412^ and C^423^ in BoNT/E1.

For the two studied BoNTs, the conformations of the open state were prepared from the cleaved 3BTA structure and the conformations of the closed state using the cleaved 3FFZ structure. These starting points of the simulations were refined using Modeller [52,53], in order to conduct an energy optimization of the structures and to relax any structural stress due to cleavage in the two chains. These two conformations, corresponding to the open state of BoNT/A1 and the closed state of BoNT/E1, are denoted in this work as cleaved X-ray models.

The closed A1 and open E1 conformations of BoNTs were built by homology modeling, using Modeller [52,53] and the sequence alignments obtained by T-Coffee [54] (Appendix A) The percentages of sequence identity between the two toxins were 35% for LC and 43% for HC, according to which, use of the homology approach was valid [55]. The template structures used for homology modeling were the PDB entries 3BTA (for the open state) and 3FFZ (for the closed state), cleaved as described above. An additional homology model was built for each BoNT, by imposing certain restraints (Appendix A) between the α-helix 485–496 (A1) or 465–471 (E1) in the belt region (Figure 1A) and the other domains of BoNTs, in order to stabilize the helix and, hopefully, the belt conformation. These four starting points for the simulations obtained by homology (two with and two without restraints) are denoted as trans models.

For each Modeller run, 200 conformations were generated using the modeling refinement method “very_slow” [52]. The conformation displaying the best DOPE score was selected, hydrogens were added to the models, and the flipping of side-chains was optimized using Molprobity [56]. The final models were then evaluated using Molprobity and QMEAN [57] scores (Appendix A). The quality scores of the models were similar to those obtained for the initial X-ray crystallographic structures 3BTA and 3FFZ. Surprisingly, the scores were slightly better for trans models than for X-ray cleaved models.

Starting from the previously calculated models, the protonation of residues at neutral (7.0) and acidic (4.7) was predicted using a web-based implementation of H++; a method for pKa calculations based on continuum electrostatic model [58,59]. The protonated residues are listed in Appendix A, along with the value of the midpoint pK(1/2) of the predicted titration curve. In the majority of cases, the latter can be approximated by the classical sigmoidal (Henderson–Hasselbalch) shape, in which case pK(1/2) = pKa [60]. The protonation state of each titrable residue was then assigned by H++, based on the comparison between pK(1/2) and the selected pH value. In detail, if pK(1/2) is ≥pH, the residue is considered as protonated. A detailed discussion on the accuracy and benchmarks of H++ predictions can be found in [58,59].

Using the two different protonation levels and the six previously defined systems, twelve systems were finally set up. They were named (Table 1) according to the type of BoNT (A1/E1), the conformational state (“clo” for close, “ope” for open), and the pH for which the protonation was defined (47 or 70). The suffix “r” was added in the case where restraints (Appendix A) were used during the homology modeling. The names of these systems are also used to name the corresponding trajectories.

### 4.2. Molecular Dynamics Simulations

For each previously described system, the protein was embedded in a large water box (183 × 148 × 194 Å), and chloride counter-ions were added to neutralize the net system charge. The total number of atoms was about 510,000 in each case; see Table 1 for details on the system compositions. All molecular dynamics (MD) simulations were performed using NAMD 2.13 [61], with the CHARMM (Chemistry at Harvard Macromolecular Mechanics) 36 force field [62] for proteins and the TIP3P (Transferable Inter-molecular Potential with 3 Points) model for water [63]. A cutoff of 12 Å and a switching distance of 10 Å were used for non-bonded interactions, while long-range electrostatic interactions were calculated using the Particle Mesh Ewald (PME) method [64]. The RATTLE algorithm [65] was used to keep all covalent bonds involving hydrogen atoms rigid, enabling a time step of 2 fs. At the beginning of each trajectory, the system was minimized for 20,000 steps, then heated up gradually from 0 K to 310 K over 31,000 integration steps. Finally, the system was equilibrated for 50,000 steps in the NVT ensemble at 310 K. In these first three stages, all carbon α atoms were kept fixed. Simulations were then performed in the NPT ensemble (P = 1 bar, T = 310 K), with all atoms allowed to move freely. Atomic coordinates were saved every 10 ps. Protein roto-translation motions were harmonically restrained throughout the simulations (with a scaled force constant of 10 kcal/mol), in order to avoid any rigid-body protein motion but leaving the internal dynamics unaltered. For each trajectory, 300 ns of production was recorded for a cumulative trajectory duration of 3.6 μs.

### 4.3. Analysis of Molecular Dynamics Trajectories

The analysis of BoNT conformations sampled along MD trajectories was realized using ccptraj [66] and the Python package MDAnalysis [37,38]. The solvent accessible surfaces were calculated using FreeSASA on each frame every 0.2 ns [67]. The solvent-accessible surfaces of residues were averaged over the time interval 150–300 ns for each of the six trajectories in Table 1; producing six values for each residue: S${}_{clo47}$, S${}_{clo47r}$, S${}_{clo70}$, S${}_{clo70r}$, S${}_{ope47}$, S${}_{ope70}$ for BoNT/A1, and S${}_{clo47}$, S${}_{clo70}$, S${}_{ope47}$, S${}_{ope47r}$, S${}_{ope70}$, S${}_{ope70r}$ for BoNT/E1. For each BoNT and each residue, six normalized surface values were obtained through dividing them by the residue surface S${}_{ave}$, calculated as $\frac{1}{6}\left(S_{clo47}+S_{clo47r}+S_{clo70}+S_{clo70r}+S_{ope47}+S_{ope70}\right)$ for BoNT/A1 and $\frac{1}{6}\left(S_{clo47}+S_{clo70}+S_{ope47}+S_{ope47r}+S_{ope70}+S_{ope70r}\right)$ for BoNT/E1. The obtained ratio values smaller than 0.9 (larger than 1.1) were considered to correspond to smaller (respectively, larger) accessible surfaces, with respect to S${}_{ave}$.

The bending angles of α-helices 1 and 2 (Figure 1B) in HC${}_{\mathrm{NT}}$ were determined using the VMD plugin Bendix [48] each frame every 0.2 ns. The figures containing structures were prepared using VMD [68] or PyMOL [69].

The relative variability of backbone angles ϕ and ψ in the belt domain was estimated by calculating the circular variances V(ϕ) and V(ψ). For a given angle θ, this parameter is defined as:(1)$$V\left(\theta \right)=1-\frac{1}{n}\sqrt{{\left(\sum_{i=1}^{n}\cos\theta_{i}\right)}^{2}+{\left(\sum_{i=1}^{n}\sin\theta_{i}\right)}^{2}},$$
where *n* is the number of trajectory frames considered and $\theta_{i}$ is the angle value in frame *i*.

## Figures and Tables

**Figure 2 toxins-14-00644-f002:**
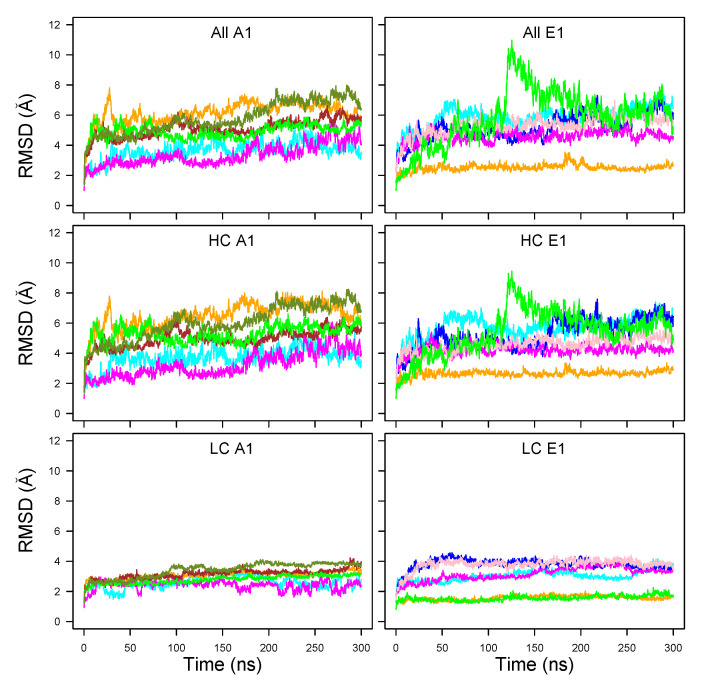
Cα root-mean-square deviation (RMSD, Å) along the MD trajectories recorded for the botulinium toxins A1 (**left** column) and E1 (**right** column). The curves are plotted with colors: Cyan (A1ope47, E1ope47), blue (E1ope47r), magenta (A1ope70,E1ope70), pink (E1ope70r), orange (A1clo47, E1clo47), brown (A1clo47r), green (A1clo70, E1clo70), olive green (A1clo70r). The RMSDs are plotted for the whole structure (**top** row), the heavy chain (HC, **middle** row), and the light chain (LC, **bottom** row).

**Figure 3 toxins-14-00644-f003:**
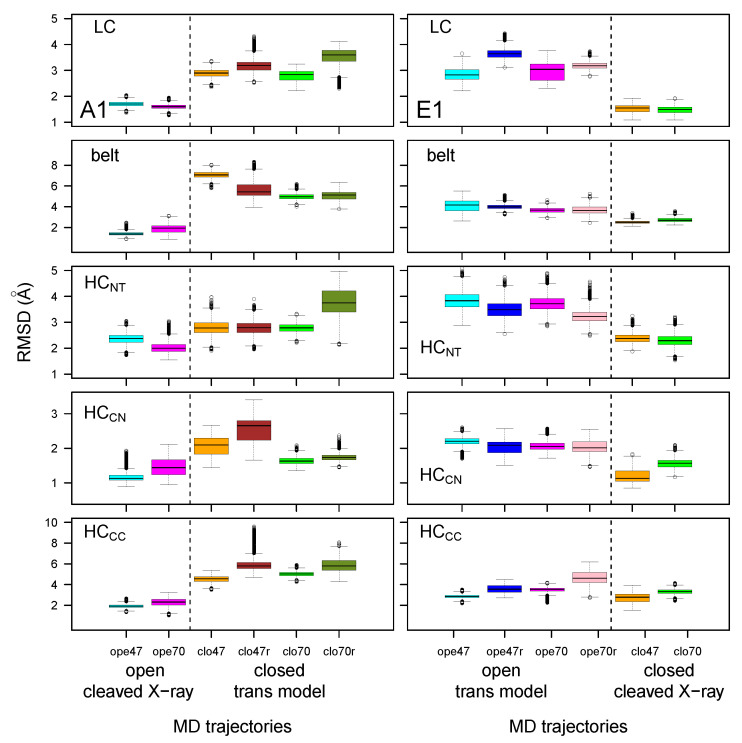
Box-and-whisker plot representation of the distributions of RMSD (Å) for domains of the botulinum toxins A1 and E1. The color code of the boxes is as follows: cyan (A1ope47, E1ope47), blue (E1ope47r), magenta (A1ope70, E1ope70), pink (E1ope70r), orange (A1clo47, E1clo47), brown (A1clo47r), green (A1clo70, E1clo70), olive green (A1clo70r). The dashed lines mark the separation between open and closed states, thus allowing us to show results relative to the various structures in the same graph, providing an overall view of the changes induced by closing/opening through the inclusion of restraints in modeling, as well as by varying the pH.

**Figure 4 toxins-14-00644-f004:**
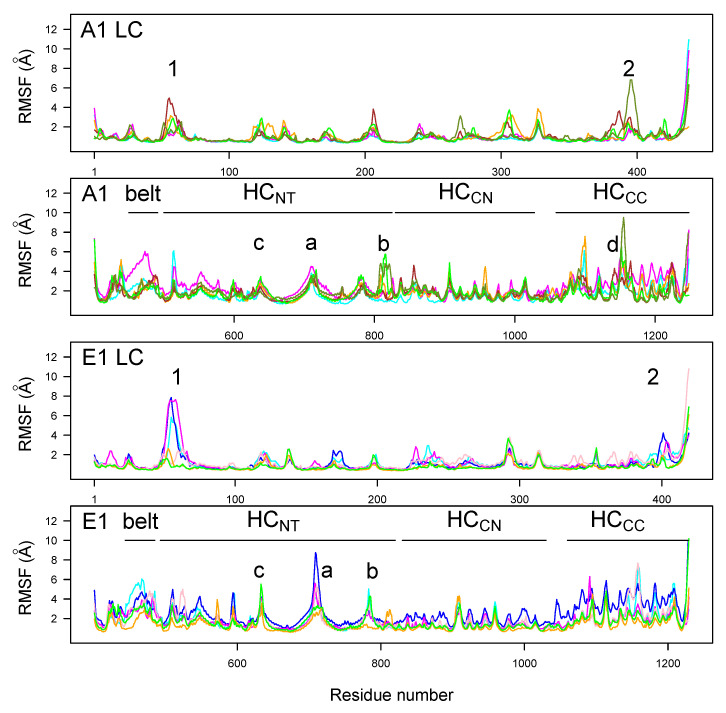
Cα root-mean-square fluctuations (RMSF, Å) calculated in the 150–300 ns interval of MD trajectories recorded for the botulinum toxins A1 and E1. The various domains (LC, HC${}_{\mathrm{NT}}$, HC${}_{\mathrm{CN}}$, and HC${}_{\mathrm{CC}}$) are indicated on the plots. The color code for the curves is as follows: cyan (A1ope47, E1ope47), blue (E1ope47r), magenta (A1ope70,E1ope70), pink (E1ope70r), orange (A1clo47, E1clo47), brown (A1clo47n), green (A1clo70, E1clo70), and olive green (A1clo70r). Peak 1: residues 63–65 (A1) and 53–60 (E1); peak 2: residues 393–394 (A1) and 392–399 (E1). Peak a: residues 746–751 (A1) and 723–734 (E1); peak b: residues 813–826 (A1) and 798–805 (E1); peak c: residues 632–656 (A1) and 601–656 (E1); peak d: residue 1188–1198 (A1).

**Figure 5 toxins-14-00644-f005:**
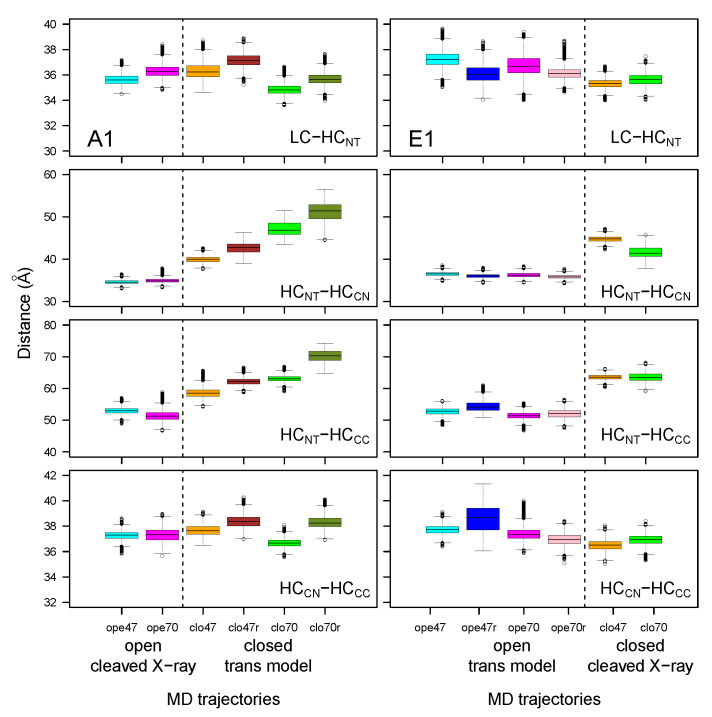
Box-and-whisker plot representation of the distributions of distances between the geometric centers of the domains within the botulinum toxins A1 and E1, averaged over the 150–300 ns interval of the MD trajectories. The color code for the boxes is as follows: cyan (A1ope47, E1ope47), blue (E1ope47r), magenta (A1ope70, E1ope70), pink (E1ope70r), orange (A1clo47, E1clo47), brown (A1clo47r), green (A1clo70, E1clo70), and olive green (A1clo70r). The dashed lines mark the separation between open and closed states; see Figure 3.

**Figure 6 toxins-14-00644-f006:**
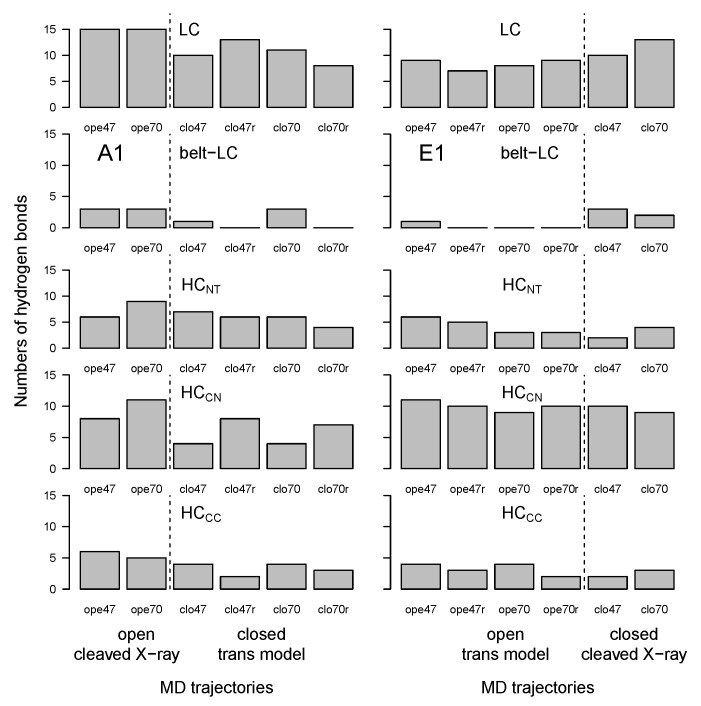
Number of hydrogen bonds between domains of the botulinum toxins A1 and E1, calculated in the 150–300 ns interval of trajectories. Long-range hydrogen bonds were detected along the trajectories by analyzing each frame every 0.2 ns, using the python package MDAnalysis [37,38]. The numbers of hydrogen bonds were calculated as those present more than 60% of the time and separated by more than 10 residues in the sequence. The dashed lines mark the separation between open and closed states; see Figure 3.

**Table 1 toxins-14-00644-t001:** System composition for molecular dynamics simulations. The names of the systems reported in the first column, are detailed in Section 4.1. “Preparation of starting conformations of toxins”. The characters “clo” and “ope” refer to the closed and open states of BoNTs, the characters “47” and “70” refer to the pH values, and the character “r” refers to the trans models determined under the distance restraints provided in Appendix A.

System	Number of Water Molecules	Counter-Ions	Total Number of Atoms
A1clo47	162,483	23 Cl-	508,335
A1clo47r	162,409	28 Cl-	508,123
A1clo70	162,514	2 Cl-	508,386
A1clo70r	162,418	5 Cl-	508,104
A1ope47	162,499	19 Cl-	508,373
A1ope70	162,531	No counter-ions	508,431
E1clo47	162,749	24 Cl-	508,408
E1clo70	162,777	2 Cl-	508,448
E1ope47	162,634	35 Cl-	508,083
E1ope47r	162,639	32 Cl-	508,092
E1ope70	162,673	3 Cl-	508,136
E1ope70r	162,668	6 Cl-	508,127

**Table 2 toxins-14-00644-t002:** Definition of domains used for BoNTs A1 and E1.

Domains	Residues Range A1	Residues Range E1
LC (catalytic)	1–438	1–419
HC	439–1286	420–1248
HC${}_{N}$	439–865	420–847
HC${}_{C}$	866–1286	848–1248
belt	484–529	464–512
belt α-helix	485–496	465–471
HC${}_{\mathrm{NT}}$	538–865	513–847
HC${}_{\mathrm{NT}}$ switch	626–655	601–631
HC${}_{\mathrm{NT}}$ α-helix 1	678–744	657–723
HC${}_{\mathrm{NT}}$ α-helix 2	756–817	737–798
HC${}_{\mathrm{NT}}$ C-terminal α-helix	850–862	831–843
HC${}_{\mathrm{CN}}$	866–1066	848–1050
HC${}_{\mathrm{CC}}$	1097–1286	1080–1248

**Table 3 toxins-14-00644-t003:** Residues displaying variations in accessible surfaces between different systems. Protonated residues quoted in Appendix A are written in bold.

System	Residues more accessible in closed than in open state
A1	LC: M30 R48 T52 I154 N205 G255 N353 I376 F419 T420
	HC: L465
	HC${}_{\mathrm{NT}}$: N560 K616 A618 D619 G644 F648 V783 N788 I791
	K796 **E799** N816 L843 I863
	HC${}_{\mathrm{CN}}$: L918
System	Residues more accessible in closed state at pH 7.0 than in other states
A1	LC: I45 G119 G120 Y180 I348 F357 N362 N402
	HC: N449
	HC${}_{\mathrm{NT}}$: I737
	HC${}_{\mathrm{CC}}$: M1124 V1176
System	Residues more accessible in closed state at pH 4.7 than in other states
A1	LC: F36 I111 P116 P156 **H170** G179 T183 A228 L232 Y233 Y251 F290
	K320
	belt: F498 G516
	HC${}_{\mathrm{NT}}$: R692 L809
	HC${}_{\mathrm{CN}}$: I922 V923 Y924 N944
	HC${}_{\mathrm{CC}}$: T1135 F1150 I1172 G1220 G1238 G1241 F1252
System	Residues more accessible in open state than in closed state
A1	LC: N5 **H39** A65 Y72 G211 I237 K375 L422
	belt: Q491 L494 N500 N509
	HC${}_{\mathrm{NT}}$: I556 L558 T559 V562 V572 T614 K710 A732 I752 I821
	HC${}_{\mathrm{CN}}$: L869 S886
	HC${}_{\mathrm{CC}}$: R1121 R1169 N1247 I1248 G1269
System	Residues more accessible in closed state than in open state
E1	LC: T234 Q237 S292 L391 R394
	belt: I489
	HC${}_{\mathrm{NT}}$: S536 L624 A667 W686 I737 **E741** T760 E761 S763 I764
	K769 N772 **E773** K775 I776 N777 I794 I800 N809 V812 L824 T828 L833
	HC${}_{\mathrm{CN}}$: **D866** K908 N985
	HC${}_{\mathrm{CC}}$: F1157
System	Residues more accessible in open state than in closed state
E1	LC: L144 T228 Q235 N392 I396 V405
	HC: E462
	belt: T503
	HC${}_{\mathrm{NT}}$: V532 I538 K728 P821 K823 N838 K839 F841 K842
	HC${}_{\mathrm{CN}}$: S867 N911 F1044
	HC${}_{\mathrm{CC}}$: I1172

## Data Availability

The data presented in this study are openly available at https://doi.org/10.5281/zenodo.6989010, accessed on 10 September 2022.

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
