# Peer review of "In Silico Conformational Features of Botulinum Toxins A1 and E1 According to Intraluminal Acidification"

_toxins, 2022, doi:10.3390/toxins14090644_

Round 1

Reviewer 1 Report

To the extent that MD simulations are useful, this is carefully modelled and analyzed. In general if some of the tables/graphs can be displayed on molecular structures, that would be preferable.

Below are specific comments.

Define “trans models” at first use.

Figure 2

Change top line, middle line bottom line

To top row

Line 264

“In the X-ray crystallographic structures of BoNTs, the regions LBL and GBS are close in the 3D space and correspond to well defined binding regions.”

Please be more precise about what is meant by “close”.

Figures 3, 5, 7

What do the black dots outside the error bars represent?

Figures 3, 5, 6, 7

Instead of dashed line to separate open, close. Would be better to have separate panels

Otherwise, explain dashed line in the figure legends

Line 295

“The ratio values smaller than 0.9 (respectively larger than 1.1) were pointed out to correspond to smaller (respectively larger) accessible surfaces than the global average surface.”

Please re-write this sentence; I am not sure what you mean here.

Figure 8

Not sure if upper panel is needed. Maybe make the lower panel larger and label some of the residues. If you do want to keep the upper panel, state the meaning of the dashed lines and make the letters of domains bigger. Right now, LC looks like LO on my PDF version.

Figure 4

Define the peaks 1,2, a,b,c in the figure legend. Why are you mixing letters and numbers here?

Line 346

Identify which of these residues are in active site and substrate binding site (if any)

Line 475

Please clarify what you mean by “ with respect to the breaking in two chains.”

Tables S1, S2

Provide the pKa values for residues in the tables. How were residues with pKa close to pH7 or 4.7  handled?

Other supplementary data

Are MD trajectories or portions of them going to be available? This would be demanding in storage space. Maybe snapshots of key transitions?

Author Response

see the attached pdf file Toxins_Answers-to-referee_Aug_13_2022_last.pdf containing the answers to the three reviewers. 

Reviewer 2 Report

The manuscript entitled “In silico conformational features of botulinum toxins A1 and E1 shed light on translocation into the neuronal cytosol” is well written and detailed. I have no suggestions for the study. I would like to suggest to the authors to include in the text some additional sentences (in the section discussion) to detail the potential impact of this study on the research in BoNTs field, considering both the use of BoNTs in the treatment of some neuro-muscular disorders and the control/prevention of natural botulism. In the text, there are several acronyms. I would like to suggest that the authors check the manuscript to be sure that all acronyms appear in the extensive form when used for the first time. 

Author Response

see the attached file Toxins_Answers-to-referee_Aug_13_2022_last.pdf containing the answers to the three reviewers.

Reviewer 3 Report

This is a well written but highly complex manuscript containing large amounts of very detailed information relating to a very specialised area of molecular dynamic simulations of BoNT molecules.  As such, a reviewer cannot check or even evaluate the complex data and methods used as provided in the text.  To the vast majority of BoNT scientists, the detail and findings will not be understood.  In addition, no clinician using BoNT for medical treatments will understand the data, purpose or the value of the results.  Indeed, this is an entirely academic manuscript of limited value in the BoNT world.

Author Response

(The authors gave the same response as above.)
